# SPR-Based Detection of ASF Virus in Cells

**DOI:** 10.3390/ijms23137463

**Published:** 2022-07-05

**Authors:** Alessandro Capo, Alessia Calabrese, Maciej Frant, Marek Walczak, Anna Szczotka-Bochniarz, Georgios Manessis, Ioannis Bossis, Maria Staiano, Sabato D’Auria, Antonio Varriale

**Affiliations:** 1Institute of Food Science, CNR Italy, 83100 Avellino, Italy; alessandro.capo@isa.cnr.it (A.C.); alessia.calabrese@isa.cnr.it (A.C.); maria.staiano@isa.cnr.it (M.S.); antonio.varriale@isa.cnr.it (A.V.); 2URT-ISA, CNR at Department of Biology, University of Naples Federico II, 80126 Napoli, Italy; 3Department of Swine Diseases, National Veterinary Research Institute (NVRI), al. Partyzantow 57, 24-100 Pulawy, Poland; maciej.frant@piwet.pulawy.pl (M.F.); marek.walczak@piwet.pulawy.pl (M.W.); anna.szczotka@piwet.pulawy.pl (A.S.-B.); 4Laboratory of Anatomy and Physiology of Farm Animals, Department of Animal Science, Agricultural University of Athens (AUA), Iera Odos 75 Str., 11855 Athens, Greece; gmanesis@aua.gr; 5Laboratory of Animal Husbandry, Department of Animal Production, School of Agriculture, Faculty of Agriculture, Forestry and Natural Environment, Aristotle University of Thessaloniki, 54124 Thessaloniki, Greece; bossisi@agro.auth.gr; 6Department of Biology, Agriculture, and Food Sciences, National Research Council of Italy (CNR-DISBA), Piazzale Aldo Moro 7, 00185 Rome, Italy

**Keywords:** African swine fever virus (ASFV), detection, antibody, indirect ELISA test, surface plasmon resonance (SPR)

## Abstract

African swine fever (ASF) is one of the most dangerous hemorrhagic infectious diseases that affect domestic and wild pigs. Currently, neither a vaccine nor effective treatments are available for this disease. As regards the degree of virulence, ASFV strains can be divided into high, moderate, or low virulence. The main detection methods are based on the use of the polymerase chain reaction (PCR). In order to prevent an uncontrolled spread of ASF, new on-site techniques that can enable the identification of an early-stage disease are needed. We have developed a specific immunological SPR-based assay for ASFV antigen detection directly in liquid samples. The developed assay allows us to detect the presence of ASFV at the dose of 10^3^ HAD_50_/_mL_.

## 1. Introduction

African swine fever (ASF) is a hemorrhagic disease affecting domestic pigs and wild boar with devastating consequences for the global swine industry and food security. For these reasons, ASF has been characterized as an emerging and notifiable disease by the World Organization for Animal Health (WOAH). The etiological agent of African swine fever is the African swine fever virus (ASFV), a 260–300 nm diameter icosahedral DNA virus comprising an envelope, capsid, inner capsule membrane, core shell, and inner core [1,2,3,4,5]. ASFV is a double-stranded linear DNA virus, and its genome is composed of 170–190 Kb, encoding 150 different proteins, such as p30, p54, p72, pp220, p17, CD2V, p10, p12 [6], and other proteins with structural function in the virus. In fact, ASFV contains at least 68 different structural proteins and 21 cellular proteins [1]. Among the structural proteins, p30 appears early during ASFV infection and it represents the most abundantly expressed viral protein in this phase [7,8]. The protein p30 is one of the most antigenic proteins among the structural proteins that compose the virion of ASFV. The protein p30, encoded by the CP204L gene, is located in the inner membrane of the viral envelope. The protein is produced before the initiation of viral DNA synthesis and the protein expression persists until the end of the viral life cycle. Consequently, a diagnosis based on the use of specific antibodies for the detection of p30 is of great importance for an early prevention and control of ASF [9,10]. In addition, p30 has been determined as a highly immunogenic protein that stimulates the highest level of antibody response during ASFV infection in pigs [11]. All these characteristics make p30 a good target for diagnostic assays to detect ASFV infection [12]. The ASFV infection, mediated by clathrin-mediated endocytosis of host macrophages, can induce high, moderate, or low virulence, as well as it being sometimes asymptomatic. In case of infection with highly virulent strains, the animal morbidity and mortality reach 100% [13].

The main route of transmission in Europe occurs by contact between infected and healthy animals, as well as between domestic pigs and wild boars. Among domestic pigs, the disease is transmitted mainly via oral and nasal routes. However, other transmission routes are possible, including via tick bites, scarifications, or injections [14]. Global swine production is threatened by ASFV spread, in the absence of an efficacious vaccine and the uncontrolled diffusion of the virus in new regions of the world, especially in the wild boar population. Actually, in the absence of either a vaccine or treatment for ASF, rapid diagnosis is crucial in disease surveillance, control, and prevention [15]. As an example, it may serve the recent ASF outbreak in China in 2018, followed by the diffusion of the disease in other Asian countries [16,17,18].

The commonly used method for ASFV detection is based on the use of the polymerase chain reaction (conventional or real-time PCR) that allows for a high sensitivity and specificity analysis. However, it is costly, time-consuming and it is limited only to well-equipped laboratories. Different isothermal molecule amplification assays, including loop-mediated isothermal amplification assay (LAMP), polymerase cross-linking spiral reaction [19], cross-priming amplification method [20], chimeric DNA/LNA-based biosensor [21], and droplet digital PCR [22], have been developed as alternative methods to the PCR-based molecule assay. The detection of ASFV with LAMP [23], for instance, shows result concordance with real-time PCR, but it requires the use of four or more primers whose design is complex [24]. An example of an early detection method is the application of recombinase polymerase amplification combined with a lateral flow strip (RPA-LFD) [25].

In this emergency, before ASF becomes well-established all over the world, the investigation of new methodologies able to easily detect ASFV in the farm is needed. In this context, biosensors could be a tool enabling us to overcome the current disadvantages in ASFV detection. Recently, in the frame of the Swinostics project (H2020 n° 771649), we have selected, identified, and characterized the molecular recognition elements (MREs) necessary to develop a rapid assay for the real-time detection of six different emergent swine respiratory viruses [26,27,28,29,30].

In the last years, surface plasmon resonance -based biosensors have been widely used as tools for characterizing and quantifying biomolecular interactions as well as for the detection of analytes associated with medical [31] and veterinary diagnostics [32].

In this work, we present the results obtained using surface plasmon resonance (SPR) measurements to characterize the binding capability of selected biological probes to the ASFV recombinant p30 protein (24.4 KDa) antigens and to the ASFV samples grown in porcine alveolar macrophage (PAM) cells. The obtained data allowed us to design a non-destructive, label-free, and real-time optical surface plasmon resonance (SPR) immunoassay for the ASFV detection. In particular, we developed an SPR-based assay to detect the presence of the p30 antigen and the virus in the supernatant of swine cell cultures infected with ASFV.

## 2. Results

To develop an SPR-based sensor for the ASFV detection, we used a commercial specific polyclonal antibody serum (ASFV11-S). After an affinity chromatography purification step, the obtained polyclonal anti-ASFV antibodies were characterized regarding their reactivity. The antibody binding capabilities towards the p30 antigen and the ASFV virus were evaluated through Western blot analysis (data previously published [26]), and an indirect ELISA test. For this purpose, we used an isolated virus suspension with 10^7^ HAD_50_/_mL_ titer. The mean Ct value of the 1:10 diluted suspension was 22.9 (±0.2). Before performing the ELISA tests, the detection limit of the virus suspension serial dilutions was evaluated by real-time PCR experiments. The obtained results are gathered in Table 1. The limit of detection of the assay was established at the dose of 10^2^ HAD_50_/_mL_.

The data reported in Figure 1 show that the polyclonal anti-ASFV antibodies are able to recognize the p30 antigen at a concentration up to 0.01 µg/mL and the ASFV Pol19/54204 sample at a concentration up to 1:10 dilution (10^6^ HAD_50_/_mL_ titer).

The polyclonal anti-ASFV antibodies were used for the development of the SPR-based immunoassay. The SPR experiments were divided into three main steps: (1) pH scouting and antibodies immobilization; (2) antibody binding capability evaluation; (3) virus sample analysis.

### 2.1. pH Scouting and Immobilization of Antibodies on the Sensor Surface

The pH scouting procedure was performed to identify the best conditions for the antibody immobilization onto a CMD 2D chip surface. As reported in the Material and Methods section, this procedure was obtained by testing different pH values at different experimental conditions such as different antibodies concentrations, different times of contact, and different flow rate values.

In Figure 2 are reported the results obtained from the combination of two different parameters (pH values and antibody concentration). In detail, Figure 2a reports the different concentrations of the tested antibodies. The data analysis shows that the best result was obtained by using a concentration of polyclonal antibody of 100 µg/mL. Figure 2b shows the different tested pH values. The MES buffer at a pH of 5.0 was chosen for the immobilization conditions because the plateau was reached early and then it was maintained for longer time than in the other tested conditions. The selected conditions guarantee a mild immobilization of antibody onto the chip’s surface in terms of the density of ligands on the surface to avoid the crowding effects. The results obtained with different times of contact and different flow rate values are not reported.

Finally, the best conditions for the immobilization procedure of the antibody were identified: 10 mM MES pH 5.0 as immobilization buffer; 200 mM EDC/50 mM NHS for the surface activation; polyclonal anti-ASFV at 100 µg/mL; 10 µL/min as flow rate; 8 min of contact time. Even if the selected antibody immobilization conditions do not represent the best values obtained in terms of mDeg (as reported in Figure 2), they were used to avoid a crowding effect on the sensor chip surface as consequence of an excess of the anti-ASFV immobilized. Figure 3 shows the results of the antibody immobilization procedure. A significant level of anti-ASFV antibodies’ immobilization was obtained. In particular, the data show a value of 312 mDeg (3.12 ng/mm^2^), which ensures that the chip was coated with an optimal number of antibodies. The obtained surface was tested for antibody binding capability using the recombinant antigen p30. It was used to evaluate the simulated spiked virus sample.

### 2.2. Analysis of the Binding Capacity of the Antibody Using the Recombinant Antigen (p30)

SPR binding measurements were carried out on the CMD 2D chip to evaluate the residual binding capability of the immobilized anti-ASFV. For this purpose, a recombinant protein p30 was used. The complex anti-ASFV-p30 kinetic properties, the optimal binding conditions, and the operational dynamic range of the chip in terms of K_D_, LOD, and Bmax were identified.

A preliminary test to evaluate the molar range of the K_D_ was performed by fluxing onto the chip’s surface a 5.0 nM solution of p30 diluted in 10 mM NaP pH 7.4 buffer. From the analysis of the data, a nanomolar value of K_D_ was identified. A complete kinetics analysis was performed testing the recombinant antigen p30 diluted in 10 mM NaP pH 7.4 buffer in a nanomolar range (from 0.01 to 200 nM). The results of the binding of p30 to the anti-ASFV are depicted in Figure 4a. The analytical signal due to the shift of resonance angle (mDeg) was measured as an average value between two reports point, one fixed at start of the sample injection (start of the association phase) and the second fixed at the end of the sample injection (start of the dissociation phase). The kinetic evaluation analyses were performed with TraceDrawer™ data analysis software. The results were fitted by a one-to-one model, and the kinetic reaction rate constants (ka and kd) and the K_D_ were estimated (Table 2). The kinetic parameters ka and kd allow us to obtain information on how fast the binding happens, while K_D_ determines the amount of complex formed at equilibrium.

The affinity constant value of the immobilized anti-ASFV to the p30 was evaluated through a titration curve (Figure 4b). The SPR signal was kept at the end of the association phase as the average between two close reports points. The response of antigen bound to the antibody was plotted against the antigen concentration, and then the results were fitted by a non-linear regression equation (Y = Bmax × c/(c + K_D_) with TraceDrawer™ data analysis software. The results are presented in Figure 4b. The obtained values of K_D_, LOD and Bmax, calculated by TraceDrawer™ data analysis software, are respectively 5.04 × 10^−7^ ± (8.24 × 10^−11^) M, 0.01 nM and 239 ± (0.03) mDeg. A calibration curve, showing concentration through calibration (CTC), was built by using the same set of collected data.

The calibration curve reported in Figure 5 was obtained by using the signal values of p30 with known concentration. The calibration curve was built through eight different p30 concentrations (Table 3). The signal was extracted at the end of the association phase as an average between two close report points. The plotted points were fitted following a Four Parameter Eq Lo-Hi model, a non-linear model that gives an increasing signal at increasing concentrations (Figure 5b). Based on the obtained calibration curve, the concentrations of a second set of measured spiked p30 samples (0.5, 5 and 50 nM) (Figure 5a) were estimated and reported in Table 3. The estimated concentration values were close to the concentration values of the p30 spiked samples assayed.

### 2.3. Regeneration Procedure

After the antibody immobilization on the chip surface, we identified a protocol for the complete regeneration of the chip surface to prevent the binding capability of the antibodies. Different procedures were analyzed. A regeneration protocol fluxing for 3 min at 30 µL/min of a solution of 10 mM glycine pH 3.0 was identified as the best procedure. This protocol assures that the signal comes back to the basal level before the sample injection, and it is possible to reuse the chip up to ten times before a reduction of the antibody binding activity is able to be noticed.

### 2.4. Analysis of the Binding Capacity of the Antibody and Identification of LOD of the Chips Using the Virus Sample in Real Matrix

After the analysis of the chip performance with the p30 antigen, we investigated the binding capability of the chip with the virus samples. Two set of the virus samples were analyzed. One set was prepared directly diluting the Pol19/54204 and the second set was spiked at known concentration values. The ASF virus sample Pol19/54204, with a 10^7^ HAD_50_/_mL_ titer provided by NVRI, was serially diluted in a range from 1:10 to 1:10,000 (from 1 × 10^6^ to 1 × 10^3^ HAD_50_/_mL_) and analyzed by the SPR-developed assay. The results are reported in Figure 6a.

By the CTC tool of TraceDrawer™ data analysis software, a calibration curve was obtained using the same set of data collected (Figure 6b,c). The calibration curve reported in Figure 6c was obtained with the signal values of virus sample Pol19/54204 with a known titer. The calibration curve was obtained through nine different virus concentrations (Table 4). The signal was extracted at the end of the association phase as an average between two reported points. The plotted points were fitted following a Four Parameter Eq Lo Hi model (a non-linear model) that gives an increasing signal with increasing concentrations (Figure 6c). Based on the obtained calibration curve, three concentrations of a second set of spiked virus samples (1:15; 1:20 and 1:800) were analyzed (Figure 6b). The results are reported in Table 4. The estimated virus titers in the second set of samples were, respectively: 6.08 × 10^5^; 3.10 × 10^5^; 6.50 × 10^4^ (HAD_50_/_mL_). Based on the obtained calibration curve, the LOD of the assay was estimated to be at the dose of 10^3^ (HAD_50_/_mL_).

## 3. Materials and Methods

### 3.1. Materials

The p30 recombinant protein (0.5 mg/mL) ref. # ASFV15-R-10 and polyclonal antibody serum (anti-ASFV) ref. # ASFV11-S were purchased from Alpha Diagnostic Intl Inc. (San Antonio, TX, USA). 2-(N-morpholino) ethanesulfonic acid (MES), 1-ethyl-(3-3-dimethylaminopro-pyl) carbodiimide hydrochloride (EDC), N-hydroxysuccinimide (NHS), 0.01 mol/L phosphate-buffered saline (PBS) buffer pH 7.4, and 0.1 mol/L sodium hydroxide (NaOH) solution pH 13.0 were purchased from Sigma Aldrich (St. Louis, Missouri, USA). rProtein A Sepharose Fast Flow ref. # 17,127,902 was purchased from Cytiva (Washington, DC, USA). CMD 2D sensor chip ref. # SPR102-CMD-2D was purchased from Bionavis™, (Tampere, Finland). A PVDF Immobilon P membrane was purchased from Merck Millipore, (Burlington, MA, USA). Amersham ECL plus was purchased from GE Healthcare (Chicago, IL, USA). ELISA plate Immuno Breakable Modules Clear, C8 LockWell, MaxiSorp was purchased from Thermo Fisher Scientific (Waltham, MA, USA). All other reagents were of the highest commercially available quality and used as received. All aqueous solutions were prepared with MillyQ water. Porcine primary pulmonary alveolar macrophages (PPAMs) were purchased from the Technical University of Denmark (Lindholm, Denmark). SteriFil syringe filters ref. # S33CA045S were purchased from Microlab Scientific (Shanghai, China). 1640 RPMI Medium ref # P04-16500 was purchased from Pan Biotech (Aidenbach, Germany). Gibco fetal bovine serum (FBS) ref. # 10,270,106 was purchased from Thermo Fisher Scientific (Waltham, MA, USA). Antibiotic–Antimycotic solution ref # A5955 was purchased from Sigma Aldrich (St. Louis, MO, USA). QIAamp DNA Mini Kit ref # 51,306 Kit was purchased from Qiagen (Hilden, Germany). Virotype ASFV PCR Kit ref # VT281905 was purchased from Qiagen (Hilden, Germany).

### 3.2. ASFV Sample Collection and Preparation

#### 3.2.1. Virus Isolation and Propagation

The virus used in this study (Pol19/54204) was isolated from the ASF wild boar outbreak in Poland no. 1977 (wild boar found dead; date of confirmation: 19.11.2019; sample location: voivodship—Lubuskie, poviat—Nowa Sól, municipality—Nowa Sól). Strain Pol19/54204 belongs to genotype II, a highly pathogenic group, identified previously as genetic group II by Mazur-Panasiuk et al. [33]. The material was collected by the local employees of the Veterinary Inspection within the frame of the ASFV monitoring program in Poland. An aliquot of 100 µL filtered (0.45 µm) spleen homogenate (diluted in PBS 1:10, *v*/*v*) was added into 24-well plates containing porcine primary pulmonary alveolar macrophages (PPAMs), suspended in 900 µL of growth medium containing 1640 RPMI Medium (PAN Biotech, Aidenbach, Germany), supplemented with 1% Antibiotic–Antimycotic (A/A) solution (Sigma-Aldrich, St. Louis, MO, USA), 10% of fetal bovine serum (Gibco, Thermo Fisher Scientific, Waltham, MA), and washed pig erythrocytes (1:300 *v*/*v*). After 7 days of incubation (37 °C, 5% CO_2_) positively heme-adsorbing cultures were intended for passage and further analyses.

#### 3.2.2. Virus Titration and Quantification

PPAMs cultures, suspended in growth medium (2.2.1) were placed into 96-well plates in quadruplicates and inoculated with the isolated virus in a 10-fold series dilution. Determination of 50% hemadsorption doses endpoint titer (HAD50/mL) in the first passage of isolated virus was calculated using Spearman-Kärber method [34].

An aliquot of 200 µL of virus suspension (diluted 1:10, PBS, *v*/*v*) (first passage) was intended for manual DNA extraction using the QIAamp DNA Mini Kit (Qiagen, Hilden, Germany). A Virotype ASFV PCR Kit (Qiagen, Hilden, Germany) was used to conduct a real-time PCR reaction, according to the manufacturer’s instructions.

The determination of the limit of detection was conducted by serial 10-fold dilutions of the ASFV isolate in nuclease-free water in four repetitions. The DNA extraction was performed with the use of the QIAamp DNA Mini Kit (Qiagen, Hilden, Germany) according to the manufacturer’s procedure. For the amplification of the DNA, real-time PCR was conducted with the use of the Virotype ASFV PCR Kit (Qiagen, Hilden, Germany), according to the manufacturer’s instructions.

### 3.3. Antibody Selection and Characterization

Polyclonal, commercially available antibodies were identified and purchased (Alpha Diagnostic Intl Inc., San Antonio, TX, USA). The first step was the purification of the antibodies from the ASFV11-S serum, then we evaluated their binding capability through Western blotting and indirect ELISA tests.

#### 3.3.1. Antibody Purification

The polyclonal antibodies were purified by affinity chromatography, using a rProtein A Sepharose 4 Fast Flow resin (Cytiva, Washington, DC, USA), from the purchased serum following the supplier specifications. In brief, 1 mL of serum was diluted 1:1 in 50 mM sodium phosphate (NaP) at pH 7.0 (binding buffer) and applied to 1 mL of rProtein A Sepharose 4 Fast Flow resin. The IgGs were purified as described by Pennacchio et al. [35]. The IgG sample was eluted with 0.1 M sodium citrate at pH 3.0 and immediately buffered with 1.0 M sodium borate at pH 9.0. The elution of IgGs proteins was monitored by absorbance at λ = 278 nm and SDS-PAGE (12% acrylamide) was carried out to evaluate the purity of the samples (data not shown). The obtained pure samples were collected and dialyzed against 10 mM PBS at pH 7.4. At the end of the purification process, polyclonal antibodies were concentrated at 2 mg/mL.

#### 3.3.2. Western Blot Experiments

A volume of 10 μL of supernatant of the swine cell culture infected with ASFV (ASFV Pol19/54204), p30 recombinant protein (20 μg/lane), and purified anti-ASFV (20 μg/lane) were separated by SDS-PAGE (15% acrylamide) and then transferred overnight at 4 °C, 75 mV, onto a 0.45 μm PVDF Immobilon P membrane. Membranes were then blocked for 1 h at room temperature in 50 mL of the TBS–T blocking buffer (TBS–Tween 0.05%, containing 5% of non-fat dried milk), were washed three times with TBS–T (10 min for each washing) and then were incubated with purified polyclonal IgG anti-ASFV (1 µg/mL), diluted in the diluting buffer (TBS–T, containing 1% of non-fat dried milk, 0.05% Tween) for 1 h at 37 °C. After three washings with TBS–T (10 min for each washing), membranes were incubated with secondary antibody (goat anti-rabbit–HRP conjugate, 1 µg/mL), diluted in the diluting buffer for 1 h at room temperature, and washed three times as described above. Finally, proteins were visualized by chemo-luminescence using the Amersham ECL plus (GE Healthcare, Chicago, IL, USA) and X-ray films were manually developed.

#### 3.3.3. ELISA Test

To determine the antibody titer, an indirect ELISA test was performed as described by El Kojok et al. [36], with slight modifications. The ELISA plate was coated with 50 μL/well of recombinant antigen (from 1 µg/mL to 0.001 µg/mL) and/or virus sample (from 1:10 to 1:1000) diluted in 0.05 M carbonate buffer at pH 9.6 and incubated overnight at 4 °C. As a negative control, some wells were coated by coating buffer. The wells were washed three times with washing buffer (TBS 0.01 M pH 7.4 containing 0.05% Tween-20; TBST) and after the incubation with 200 μL/well of blocking buffer (TBS supplied with 5% *w*/*v* non-fat dried milk), at 37 °C for 2 h, the plate was rinsed three times. After this step, 50 μL/well of polyclonal antibodies, anti-ASFV (1 μg/mL) diluted in TBS, 1% non-fat dried milk, and 0.05% *v*/*v* TWEEN 20 buffer, were incubated at 37 °C for 2 h. The plate was rinsed (three times), 50 μL/well of goat anti-rabbit IgG-HRP antibody (0.5 μg/mL), was added, and the wells were incubated for 1 h at 37 °C. Finally, the enzyme substrate solution (TMB) was added (100 μL/well), and the wells were incubated at 37 °C, then the color development was quenched by adding stopping solution HCl (2.5 M 50 μL/well) after 10 min. A Tecan Infinity 200 Pro (Tecan, Männedorf, Switzerland) micro-plate reader was used to measure the absorbance at 450 nm.

### 3.4. Surface Plasmon Resonance (SPR)

The purified polyclonal antibodies of anti-ASFV were immobilized on a CMD 2D sensor chip, which has a dextran-layered gold surface. The SPR detects and measures changes in the refractive index as a consequence of the binding and dissociation of molecules. The change in refractive index is proportional to the quantity (mass) of analyte (p30 or ASFV) interacting with the ligand (anti-ASFV). Similarly, the interaction between the antibody and virus particle causes a shift in the angle of refraction, and the SPR phenomenon occurs. The signal is expressed in the degree of angle (mDeg), and these shifts, monitored continuously over time, are recorded as a sensorgram. A response of 0.1° in the response angle represents a change in surface protein concentration of about 1 ng/mm^2^. The SPR measurements were carried out on the MP-SPR Navi™ 210A VASA instrument (Bionavis™, Tampere, Finland) by using the CMD 2D sensor chip. All experiments were performed three times, at a flow rate of 10 μL/min, at room temperature, with an HBS–EP buffer. The data acquired were processed with an MP-SPR Navi™ DataViewer (Bionavis™, Tampere, Finland) and analyzed using TraceDrawer™ (TraceDrawer, Uppsala, Sweden) data analysis software.

#### 3.4.1. pH Scouting

Before the immobilization procedure, in order to determine the best immobilization condition of anti-ASFV on the CMD 2D sensor’s surface, a pH scouting procedure was performed with an MP-SPR Navi™ 210A VASA instrument. For this purpose, we tested different pH conditions, concentrations of antibodies, times of contact and flow rates. In 10 mM sodium acetate or 5 mM MES at pH 3.5, 4.0, 4.5, 5.0 and 5.5, the selected polyclonal anti-ASFV antibodies were diluted to a final concentration of 10, 20 and 25, 50 and 100 μg/mL, respectively. The flow rate tested was 10 and 25 μL/min and contact times ranging from 5 to 30 min were explored. At the end of each injection, a washing solution (100 mM sodium hydroxide) was injected to remove any unbound molecules from the chips’ surface. From the sensorgram analysis, the best conditions for the immobilization procedure were chosen.

#### 3.4.2. Surface Preparation (Polyclonal Anti-ASFV Immobilization)

The immobilization procedure of the anti-ASFV on the sensor surface is divided into several phases: the cleaning and pre-conditioning of the chip surface; activation by EDC/NHS of the carboxylic residues present on the chip’s surface; injection of the antibody (immobilization step); inactivation of the activated residues that have not bound the antibody; final cleaning of the chip. The CMD 2D sensor chip was washed two times (10 min each) with a solution of 1 M sodium chloride (NaCl)/100 mM sodium hydroxide (NaOH). After, for the surface activation, the carboxy-methylated dextran surface was fluxed with a 1:1 mixture of 200 mM EDC/50 mM NHS pH 7.0 in the flow cell 1 and 2. After the activation step, the polyclonal antibodies diluted in 10 mM MES pH 5.0 at 100 µg/mL were injected only into flow cell 1 (channel 1) of the chip, while on flow cell 2 (channel 2) a 10 mM MES pH 5.0 buffer was fluxed. The injection step was performed with a flow rate of 10 μL/min for 8 min. After the immobilization step, the remaining NHS esters were blocked by the injection of a 1.0 M ethanolamine hydrochloride solution at pH 8.5 for 10 min with a flow rate of 10 μL/min. At the end, to remove all unbound molecules from the surface, the chip was washed two time (10 min each) with a solution of 1 M NaCl/100 mM NaOH.

#### 3.4.3. Binding Experiments

SPR measurements were carried out on the chip prepared as described above; channel 1 was used as the sensing channel because it was functionalized with polyclonal anti-ASFV, and channel 2 was used as a reference channel. All measurements were performed by the MP-SPR Navi™ 210 VASA, the sensorgrams were acquired by the MP-SPR Navi™ Control software, and the raw files acquired were analyzed by the MP-SPR Navi™ DataViewer software to obtain the arithmetical difference between channel 1 and 2. The measures were acquired at 25 °C, with a flow rate of 10 μL/min and an injection of 150 μL of the sample p30 or ASFV Pol19/54204 diluted in 10 mM NaP pH 7.4 buffer on both channels. As a running buffer, we used 10 mM Hepes buffer (pH 7.4). The analytical signal of interest was given by the shift in resonance angle (mDeg), measured as an average between two reported point, one fixed at 20 sec and the second fixed at 10 sec before the end of the sample injection, which represents the plateau or equilibrium phase. Titration curves were obtained by evaluating the binding affinity of the immobilized anti-ASFV to the recombinant antigens, p30 diluted in 10 mM NaP pH 7.4 buffer in a nanomolar range. The response of antigen bound to the MRE was plotted against the antigen concentration, and then the results were fitted by a non-linear equation with TraceDrawer^TM^ data analysis software.

#### 3.4.4. Regeneration Procedure Optimization

In order to be able to reuse the chips produced, a surface regeneration procedure has been identified which involves detaching the recombinant antigen or the virus sample from the antibody in order to make it available for subsequent measurement. Several regeneration protocols were analyzed, using different strategies (buffer) of regeneration as reported by Andersson et al. [37,38]. The protocol identified involves the use of a 10 mM pH 3.0 glycine buffer fluxed for 3 min at 10 μL/min.

#### 3.4.5. Virus Sample Experiments

Virus samples: ASFV Pol19/54204 were analyzed by the assay format optimized above and utilized to build the calibration curve. All SPR measurements were performed three times, and the obtained results were analyzed by TraceDrawer™ data analysis software. Briefly, 150 µL of virus samples were filtered with a 0.45 µm PES syringe filter and diluted in 10 mM NaP pH 7.4 buffer from 1:10 up to 1:10,000. Following the regeneration procedure previously described, each chip was tested and reused at least 10 times.

### 3.5. Statistical Analysis

Raw data were processed and analyzed through the use of different software. The SPR data acquired with MP-SPR Navi™ DataViewer and processed by the TraceDrawer™ data analysis software or Origin Pro 8 (producer, country of origin). For the ELISA test, each measurement was performed in triplicate. From the value of the triplicates the mean and standard deviation were calculated. From the mean values the mean blank values were subtracted and values of the error bar were calculated from the standard deviation. The graphs were prepared in Excel 2019 (Microsoft^®^).

## 4. Discussion and Conclusions

ASFV is the etiological agent of the ASF, affecting domestic pigs and wild boar and having a significantly negative effect on the global swine industry and food security. The average time from symptom onset to clinical outcome in ASF is 6–20 days. Laboratory testing with TaqMan real-time PCR is being widely used. However, it requires sophisticated instruments, complex sample treatment procedures, and several hours to obtain the results. Preserving the same sensitivity and specificity, some improvements in terms of instrumentation and reduction of the time of analysis were reached by LAMP, RPA [23], chimeric DNA/LNA-based biosensors [21], and droplet digital PCR [20]. However, these detection assays were based on an expensive scanner device. New improvements have come from a technique referred to as recombinase polymerase amplification (RPA) combined with lateral flow dipstick (LFD) (RPA–LFD) [39] that reduces the time of analysis and the use of very specific equipment, but, on the contrary, the sensitivity of the assay is 100-fold lower than genomic amplification. Once the diagnosis and official proceedings of suspected ASF pigs are delayed, the risk of ASF exposure and spread to other pig farms increases [40].

The biological properties of recently circulating variants in Europe and the Asia genotype II of ASFV has been widely described before [41]. Most of the field-isolated ASFV strains belong to highly or moderately virulent strains. Despite the form of the disease differing depending on the dosage, route of infection, and host’s response, viral load in blood and tissues remains high (ranging from 10^6^ to 10^8^ HAD_50_/_mL_), which is suitable to be detected by SPR [42,43,44]. However, the usefulness of SPR should be also confirmed with the use of ASFV strains of various virulence. In the case of infection with attenuated strains, detection of the disease may require serological surveillance for the detection of specific anti-ASFV antibodies.

Here, we have reported the design and develop a surface plasmon resonance immunoassay for the rapid detection of ASFV. The assay is able to detect the p30 antigen in cell culture samples that contain incomplete ASFV virions and cell lysates/fragments.

The obtained results show that the developed method is a promising alternative to the traditional methods of analysis. In fact, the assay needs only 20 min, and the data are acquired in real-time. In addition, the sample pre-treatment is not required, and the LOD of the assay is estimated at the dose of 10^3^ (HAD_50_/_mL_). Based on these features, we could envisage to realize a point-of-care device for the ASFV on-site detection.

Further studies are needed before to proceed with the development of an on-field application. The preliminary results obtained on the ASFV confirmed the possible application of SPR-based sensors for virus detection. In our case, to definitively confirm the performance of the assay, validation experiments on real swine samples (oral fluid, blood, or fecal samples) will be performed.

## Figures and Tables

**Figure 1 ijms-23-07463-f001:**
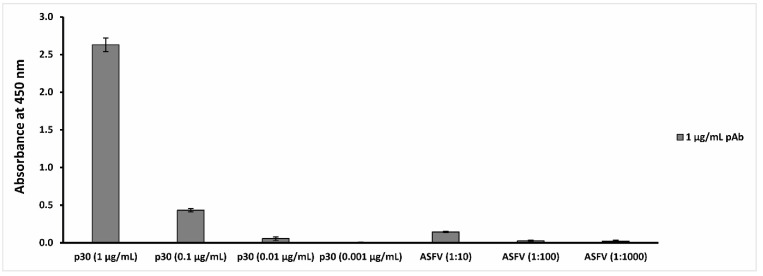
Anti-ASFV antibody binding capability evaluation. The figure reports the analysis of the results obtained by the indirect ELISA tests, in which p30 antigen was recognized up to 0.01 µg/mL, while the ASFV Pol19/54204 sample was recognized up to a 1:10 dilution.

**Figure 2 ijms-23-07463-f002:**
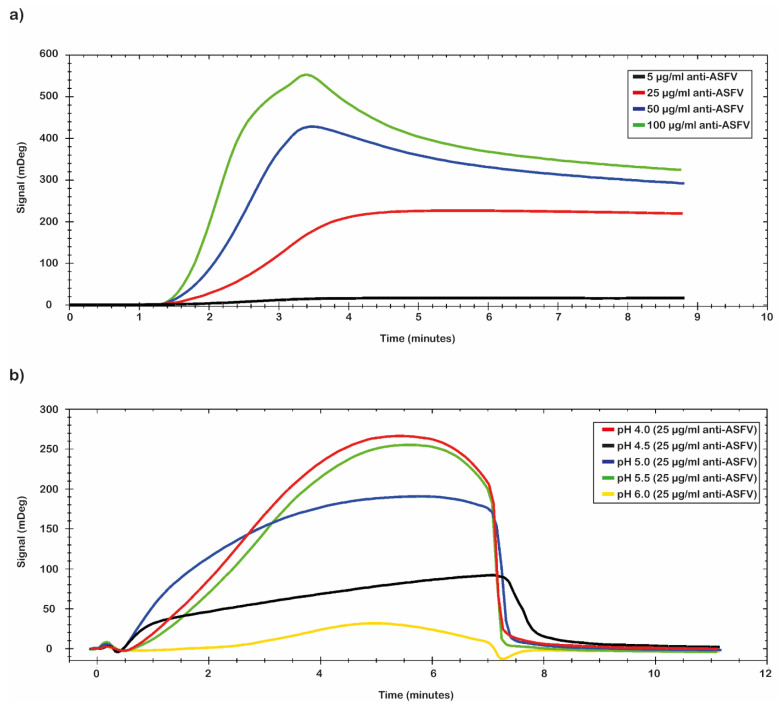
Pre-concentration analysis. Sensorgrams of pre-concentration scouting analysis; (**a**) the best value was obtained with the 100 µg/mL of polyclonal anti-ASFV, and the pH scouting analysis (green); (**b**) the optimal conditions of binding were obtained with the buffer at pH 5.0 (blue).

**Figure 3 ijms-23-07463-f003:**
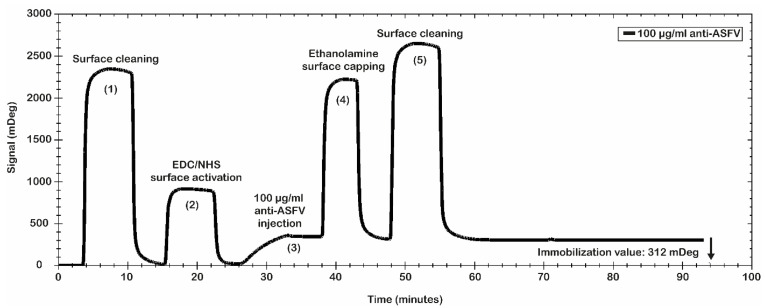
Anti-ASFV immobilization procedure. Sensorgram of amine coupling immobilization procedure. Main steps are shown: (1) cleaning, (2) activation with EDC/NHS mixture, (3) injection and bound of polyclonal antibodies, (4) ester deactivations by ethanolamine capping, and (5) cleaning. Polyclonal anti-ASFVs were immobilized at 312 mDeg.

**Figure 4 ijms-23-07463-f004:**
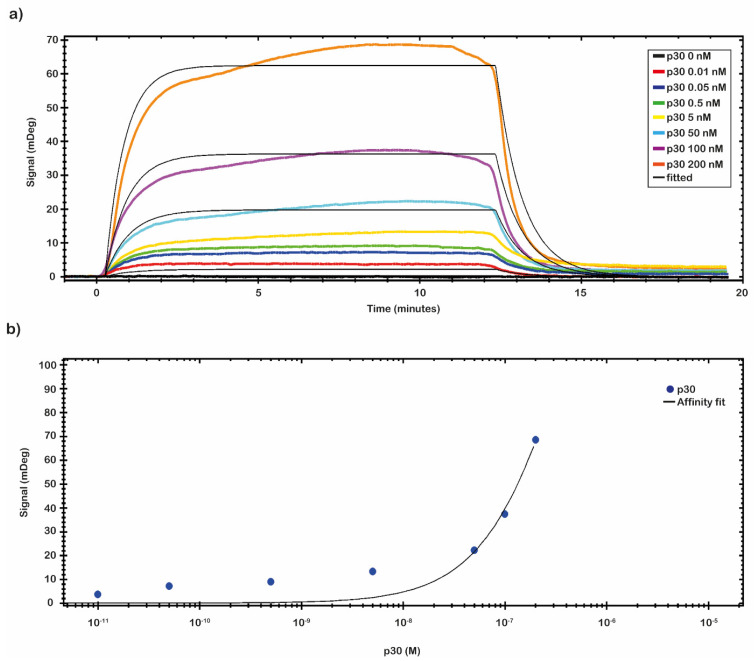
SPR assay setting up, kinetics and affinity analysis. (**a**) A complete kinetic analysis was performed with the recombinant antigen p30 in a nanomolar range (from 0.01 to 200 nM). Sensorgrams shows the binding of recombinant p30 pRec protein to anti-ASFV covalently immobilized on the sensor chip surface. All measurements were performed in 10 mM NaP pH 7.4 buffer at room temperature. (**b**) An affinity plot was obtained from the same set of collected data. The blue dots represent the values of the p30 pREC titration experiment performed by the SPR-based sensing system. mDeg values are plotted versus pRec concentration; the black curve shows a result of a non-linear fitting operation. The obtained K_D_, LOD, and Bmax values, calculated by TraceDrawer™ data analysis software, were respectively 5.04 × 10^−7^ ± (8.24 × 10^−11^) M, 0.01 nM, and 239 ± (0.03) mDeg.

**Figure 5 ijms-23-07463-f005:**
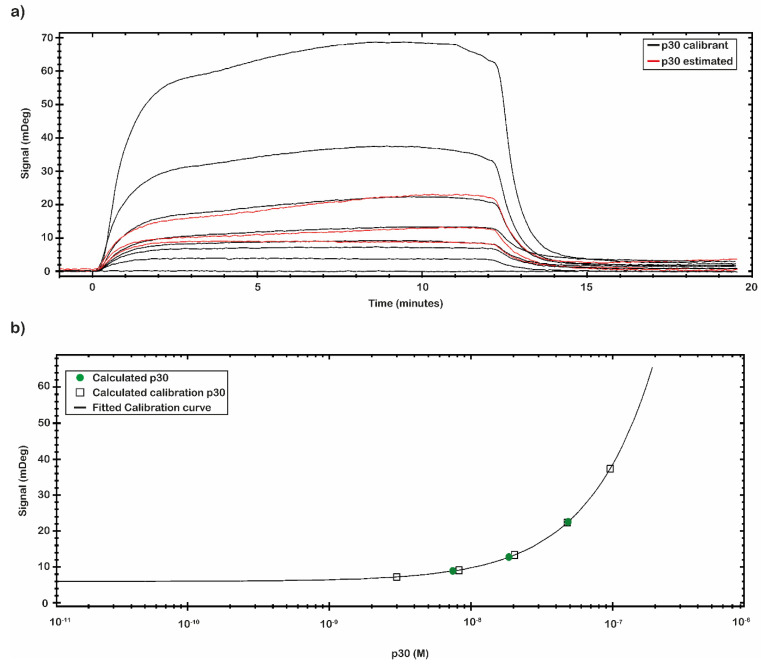
Concentration through calibration (CTC) analysis on p30 samples. (**a**) CTC analysis was performed with the recombinant antigens p30 in a nanomolar range from 0.01 to 200 nM (black line) that was used as calibrant, while another set of p30 samples was estimated (red line). All measurements were performed in 10 mM NaP pH 7.4 buffer at room temperature. (**b**) The mDeg values were plotted against the p30 calibrant sample concentration (black square) and fitted by a Four Parameter Eq Lo-Hi model. The obtained calibration curve (black line) was used to estimate the concentration of the second set of p30 analyzed samples (green circle).

**Figure 6 ijms-23-07463-f006:**
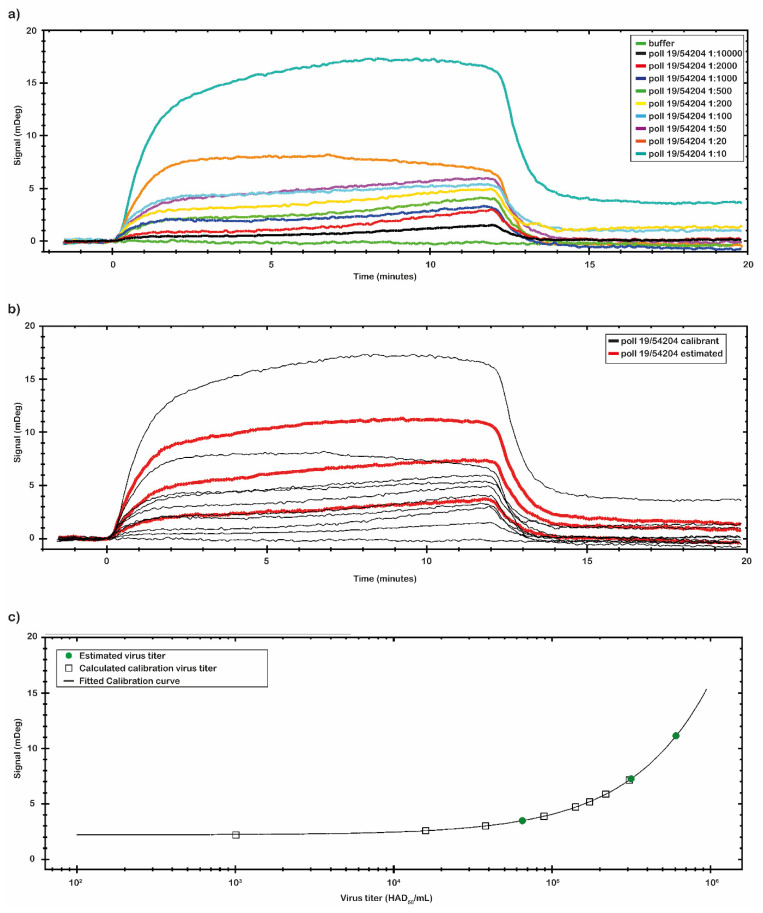
SPR experiments with ASFV sample. (**a**) Concentration through calibration (CTC) analysis on ASFV Pol19/54204 samples. (**b**) CTC analysis was performed with the ASFV sample in a range from 0 to 1 × 10^6^ (HAD_50_/_mL_) (black line) that was used as the calibrant, while another set of ASFV Pol19/54204 samples was estimated (red line). All measurements were performed in 10 mM NaP pH 7.4 buffer at room temperature. (**c**) The mDeg values were plotted against the ASFV calibrant sample concentrations (black square) and fitted by a Four Parameter Eq Lo Hi model. The obtained calibration curve (black line) was used to estimate the concentration of the second set of ASFV samples analyzed (green circle).

**Table 1 ijms-23-07463-t001:** The analysis of detection limit of ASFV isolate serial dilutions in real-time PCR.

HAD_50_ *	Ct ** Values (Replicates)				Mean Ct **	Result InterPretation
	I	II	III	IV		
10^7^	20.16	20.00	20.41	19.15	19.93	Positive
10^6^	22.77	23.12	23.11	22.66	22.92	Positive
10^5^	26.00	24.86	24.37	24.53	24.94	Positive
10^4^	28.49	27.95	28.35	27.76	28.14	Positive
10^3^	32.64	32.02	31.88	31.44	32.00	Positive
10^2^	36.37	36.57	36.08	36.27	36.32	Positive
10^1^	NoCt	NoCt	NoCt	NoCt	NoCt	Negative

* 50% hemadsorption doses endpoint titer; ** cycle threshold.

**Table 2 ijms-23-07463-t002:** Kinetic analysis results of the complex anti-ASFV-p30.

Curve Name	Bmax ([Signal (mDeg)])	ka (1/(M × s))	kd (1/s)	K_D_ (M)	Chi2 ([Signal (mDeg)]^2^)
p30 0.01 nM fitted	223.59	3.93 × 10^4^	2.03 × 10^−2^	5.17 × 10^−7^	19.49
p30 0.05 nM fitted	223.59	3.93 × 10^4^	2.03 × 10^−2^	5.17 × 10^−7^	19.49
p30 0.50 nM fitted	223.59	3.93 × 10^4^	2.03 × 10^−2^	5.17 × 10^−7^	19.49
p30 5.00 nM fitted	223.59	3.93 × 10^4^	2.03 × 10^−2^	5.17 × 10^−7^	19.49
p30 50.0 nM fitted	223.59	3.93 × 10^4^	2.03 × 10^−2^	5.17 × 10^−7^	19.49
p30 100 nM fitted	223.59	3.93 × 10^4^	2.03 × 10^−2^	5.17 × 10^−7^	19.49
p30 200 nM fitted	223.59	3.93 × 10^4^	2.03 × 10^−2^	5.17 × 10^−7^	19.49

Bmax = Maximum binding capacity of the sensor surface; Ka = Association rate constant; Kd = Dissociation rate constant; K_D_ = Equilibrium dissociation constant; Chi2 = χ^2^ value.

**Table 3 ijms-23-07463-t003:** Results of the concentration through calibration (CTC) analysis on p30 samples.

Curve Name	Calibrant	Real Concentration (M)	Calculated Concentration (M)
p30 0.00 nM (1)	Yes	0.00	
p30 0.01 nM (1)	Yes	1.00 × 10^−11^	
p30 0.05 nM (1)	Yes	5.00 × 10^−11^	
p30 0.50 nM (1)	Yes	5.00 × 10^−10^	
p30 5.00 nM (1)	Yes	5.00 × 10^−9^	
p30 50.0 nM (1)	Yes	5.00 × 10^−8^	
p30 100 nM (1)	Yes	1.00 × 10^−7^	
p30 200 nM (1)	Yes	2.00 × 10^−7^	
p30 0.50 nM (2)	No	5.00 × 10^−10^	7.48 × 10^−9^
p30 5.00 nM (2)	No	5.00 × 10^−9^	1.86 × 10^−8^
p30 50.0 nM (2)	No	5.00 × 10^−8^	4.86 × 10^−8^

(1) = Calibrant, (2) = Estimated.

**Table 4 ijms-23-07463-t004:** Results of the concentration through calibration (CTC) analysis on ASFV Pol19/54204 samples.

Curve Name	Calibrant	Real Concentration (M)	Calculated Concentration (M)
Buffer	Yes	0	
Pol19/54204 1:10,000 (1)	Yes	1 × 10^3^	
Pol19/54204 1:2000 (1)	Yes	5 × 10^3^	
Pol19/54204 1:1000 (1)	Yes	1 × 10^4^	
Pol19/54204 1:500 (1)	Yes	2 × 10^4^	
Pol19/54204 1:200 (1)	Yes	5 × 10^4^	
Pol19/54204 1:100 (1)	Yes	1 × 10^5^	-
Pol19/54204 1:50 (1)	Yes	2 × 10^5^	
Pol19/54204 1:20 (1)	Yes	5 × 10^5^	
Pol19/54204 1:10 (1)	Yes	1 × 10^6^	
Pol19/54204 1:15 (2)	No	6.6 × 10^5^	6.08 × 10^5^
Pol19/54204 1:20 (2)	No	5 × 10^5^	3.10 × 10^5^
Pol19/54204 1:800 (2)	No	12.5 × 10^4^	6.50 × 10^4^

(1) = Calibrant, (2) = Estimated.

## Data Availability

Not applicable.

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
