# Peer review of "SPR-Based Detection of ASF Virus in Cells"

_ijms, 2022, doi:10.3390/ijms23137463_

Round 1
Reviewer 1 Report
The present study aims at understanding whether African swine fever virus (ASFV) causes African swine fever (ASF), a hemorrhagic disease affecting domestic pigs and wild swine, The authors, with this work, intend to develop a methodology that detects early and with simplicity the infection in early stages, which would be a very important advance to avoid the expansion of infection among this type of animals and the economic and public health consequences of not having screening techniques and early diagnosis.
It is a good planned study. The information about the material and method are adequate. The authors provide proper results and discussion section. The paper is written concisely and clearly
The results presented are important from the point of view of Public Health, with very clarifying graphs and tables, and the conclusions are adequate to the results obtained, although more studies are needed (already with animals) to assess the usefulness of the authors' proposal.
Author Response
Response to Reviewer 1 Comments
Point 1: The present study aims at understanding whether African swine fever virus (ASFV) causes African swine fever (ASF), a hemorrhagic disease affecting domestic pigs and wild swine, The authors, with this work, intend to develop a methodology that detects early and with simplicity the infection in early stages, which would be a very important advance to avoid the expansion of infection among this type of animals and the economic and public health consequences of not having screening techniques and early diagnosis. It is a good planned study. The information about the material and method are adequate. The authors provide proper results and discussion section. The paper is written concisely and clearlyThe results presented are important from the point of view of Public Health, with very clarifying graphs and tables, and the conclusions are adequate to the results obtained, although more studies are needed (already with animals) to assess the usefulness of the authors' proposal.
Response 1: I would like to thank the reviewer for the positive evaluation of the manuscript.

Reviewer 2 Report
In this manuscript, the authors developed and presented a novel SPR-based detection method which could be used in the pig farm for early diagnosis of ASFV. ASF is a global problem, and this study could contribute of great importance. However, there is a limitation in this manuscript, and I recommended some additional experiment for improving this study.
1. The title needs to be changed. It should be contained only "SPR-based detection of virus in cells". The title presented by the authors is too broad in scope.
2. Line 22-23: The pathogenicity of ASFV is classified as high, moderate, or low. not medium.
3. Line 27: level --> dose
4. Line 35: OIE is "Organisation for Animal Health", and the official name of the organization has recently changed, so please check it again.
5. Line 57: sick --> infected
6. Line 66: Reference should be added for cases of ASFV found in other Asian regions other than China (Korea, Vietnam, Philippines, etc.).
7. Line 83: What is H2020 N771649 ?
8. Line 93: PAM cells mean "porcine alveolar macrophages cells". Please delete "Porcine" in the line 93.
9. There are too many paragraphs in the Introduction. I suggested to modify the paragraph structure to fit the flow and reduce the paragraph to 3 to 5 for easy reading.
10. Method Section : It is necessary to add more information about the pathogenicity and genotype of the virus (Pol19/54204) used in the experiment. It is very important information, because there is many types of ASFV worldwide.
11. Only one virus was used in this study. However, ASFV has several genotypes (24 genotypes) and virulence forms (peracute, acute, moderate, chronic). Only one virus (pol19/54204) was targeted in this study, and this is a significant limitation. Please described the limitation of the study in the Discussion section.
12. The sensitivity and specificity of the SPR-based ASFV diagnostic method in this study are insufficient. The diagnosis method should be differentiated ASFV from PRRSV, PCV2, or other major swine viral pathogens. Therefore, authors must conduct more experiment with the developed diagnostic method which is not amplified by PRRSV, PCV2, and others.
13. The results of this study directly targeted ASFV isolation cultured in PAM cells. The authors selected p30 for early diagnosis (according to Introduction section), the study should conduct an additional experiment which could detect ASFV in pigs' biological samples (blood, oral fluid, or nasal swab). This is very important for this diagnostic method could be applied in filed by veterinarians.
14. ASFV diagnosis by the SPR method has been published recently (https://doi.org/10.3390/bios12040213). What is the novelity which different from the recent previous study?
Author Response
Response to Reviewer 2 Comments
In this manuscript, the authors developed and presented a novel SPR-based detection method which could be used in the pig farm for early diagnosis of ASFV. ASF is a global problem, and this study could contribute of great importance. However, there is a limitation in this manuscript, and I recommended some additional experiment for improving this study.
Point 1: The title needs to be changed. It should be contained only "SPR-based detection of virus in cells". The title presented by the authors is too broad in scope.
Response 1: According to the reviewer’s suggestion, the new title of the manuscript is “SPR-based detection of ASF virus in cells”
Point 2: Line 22-23: The pathogenicity of ASFV is classified as high, moderate, or low. not medium.
Response 2: According to the reviewer’s suggestion, we have modified the text in the revised manuscript.
Point 3: Line 27: level --> dose
Response 3: According to the reviewer’s suggestion, we have modified the text in the revised manuscript.
Point 4: Line 35: OIE is "Organisation for Animal Health", and the official name of the organization has recently changed, so please check it again.
Response 4: According to the reviewer’s suggestion, we have changed the acronym from OIE to WOAH.
Point 5: Line 57: sick --> infected
Response 5: According to the reviewer’s suggestion, we have modified the text in the revised manuscript.
Point 6: Line 66: Reference should be added for cases of ASFV found in other Asian regions other than China (Korea, Vietnam, Philippines, etc.).
Response 6: I would like to thank to the reviewer for the important comment. In the revised version of the manuscript, we added the reference: Mighell, E.; Ward, M.P. African Swine Fever spread across Asia, 2018-2019. Transbound Emerg Dis 2021 68(5), 2722-2732, doi:10.1111/tbed.14039.
Point 7: Line 83: What is H2020 N771649?
Response 7: The word “H2020 N 771649” is the code of the European project that is funding this research activity.
Point 8: Line 93: PAM cells mean "porcine alveolar macrophages cells". Please delete "Porcine" in the line 93.
Response 8: According to the reviewer’s suggestion, in the revised version of the manuscript we modified it.
Point 9: There are too many paragraphs in the Introduction. I suggested to modify the paragraph structure to fit the flow and reduce the paragraph to 3 to 5 for easy reading.
Response 9: According to the reviewer’s suggestion, in the revised version of the manuscript we reduced the introduction section to five paragraphs.
Point 10: Method Section: It is necessary to add more information about the pathogenicity and genotype of the virus (Pol19/54204) used in the experiment. It is very important information, because there is many types of ASFV worldwide.
Response 10: I would like to thank to the reviewer for this comment. In the revised version of the manuscript, we added the description to the M&M section 3.2.1.
“The virus used in this study (Pol19/54204) was isolated from the ASF wild boar outbreak in Poland no. 1977 (wild boar found dead; date of confirmation: 19.11.2019; sample location: voivodship - Lubuskie, poviat - Nowa Sól, municipality - Nowa Sól). Strain Pol19/54204 belongs to genotype II, a highly pathogenic group, identified previously as genetic group II by Mazur-Panasiuk et al.”
Point 11: Only one virus was used in this study. However, ASFV has several genotypes (24 genotypes) and virulence forms (peracute, acute, moderate, chronic). Only one virus (pol19/54204) was targeted in this study, and this is a significant limitation. Please described the limitation of the study in the Discussion section.
Response 11: I would like to thank to the reviewer. According to our best knowledge, genotype II of ASFV is responsible for the current ASF epidemic since 2007. The majority of field-isolated, genotype II strains present high pathogenicity in pigs and wild boars. However, the form of the disease may differ depending on dose, route of infection, and response of the host.
Nevertheless, the most important element is the viral load in blood and tissues that is equally high reaching 106-108 HAD50/ml, and consequently suitable for detection by the SPR.
The ASFV strain used in this study was isolated from a wild boar case of the disease. It represents genotype II which is responsible for the current ASF epidemic and currently it circulates in Europe and Asia. Therefore, it could be considered representative and adequate for the present diagnostic approaches.
In the revised manuscript, we added the following sentence to the description to Discussion and Conclusion section: “The biological properties of recently circulating in Europe and Asia genotype II of ASFV has been widely described before [41].”
Most of the field-isolated ASFV strains belongs to highly or moderately virulent strains. Despite the form of the disease may differ depending on dose, route of infection and response of host, viral load in blood and tissues remains high (ranging from 106 to 108 HAD50/ml) which is suitable to be detected by SPR [42-44]. However, the usefulness of SPR should be also confirmed with the use of ASFV strains of various virulence. In case of attenuated strains infection, detection of the disease may require serological surveillance for detection of specific anti-ASFV antibodies.”
Point 12: The sensitivity and specificity of the SPR-based ASFV diagnostic method in this study are insufficient. The diagnosis method should be differentiated ASFV from PRRSV, PCV2, or other major swine viral pathogens. Therefore, authors must conduct more experiment with the developed diagnostic method which is not amplified by PRRSV, PCV2, and others.
Response 12: I would like to thank to the reviewer, to allow us to clarify this point. At the moment, no government regulation regarding the sensitivity of the ASFV detection is reported. In any case, the sensitivity of our assay (103 HAD50/ml) is not far from the gold standard method (102 HAD50/ml with PCR) used in the ASFV detection, with the advantages of virus detection without any sample pre-treatment and in less time.
In addition, this work aims to develop an SPR assay for ASFV and the obtained results can be considered as proof of concept, for future developments of a multi-detection assay for swine respiratory virus.
Point 13: The results of this study directly targeted ASFV isolation cultured in PAM cells. The authors selected p30 for early diagnosis (according to Introduction section), the study should conduct an additional experiment which could detect ASFV in pigs' biological samples (blood, oral fluid, or nasal swab). This is very important for this diagnostic method could be applied in filed by veterinarians.
Response 13: I would like to thank to the reviewer, for this comment. According to our best knowledge, the SPR sensors are suitable to confirm the presence of pathogens even when virions are fragmented. Therefore, blood should be a suitable matrix for detection of ASFV. However, these results are preliminary and a study for the detection of ASFV in field samples will be performed.
Point 14: ASFV diagnosis by the SPR method has been published recently (https://doi.org/10.3390/bios12040213). What is the novelity which different from the recent previous study?
Response 14: Thanks to the reviewer to highlight the new paper regards the ASFV diagnosis. The SPR assay developed and presented in this manuscript is different respect the assay reported in this article, in which the LAMP assay is coupled with SPR measurements.
In our work, we present a direct SPR assay that does not need any pre-treatment of the sample and it allows to detect the virus and/or virus fragments in a real sample.

Round 2
Reviewer 2 Report
Thank you for the revised manuscript for my all suggestion.
But there is still some changes for English and no mention about the limitation in Discussion section.
(there is no line number)
1. Full name of SPR should be described at the page 1 of 18 "In the last year, SPR-based biosensors ...."
2. The Table 1. "the dose of 102 (HAD50/ml)" --> "the dose of 102 HAD50/ml"
3. In page 11 of 18, only one word "Virotype" was italic. check it please.
4. In second line of Discussion and Conclusion section, wild swine --> wild boar
5. Still, I think that overall results are preliminary and difficult to apply in the field. Because the experiment was conducted 1) on cells only, and 2) it was not confirmed that the method could be distinguish other pathogens (PRRS, PCV2 etc.), and 3) only one virus was tested. In the discussion section, it should be described that these limitations (preliminary studies) and studies for future field application are needed.
Author Response
Response to Reviewer 2 Comments
Thank you for the revised manuscript for my all suggestion.
But there is still some changes for English and no mention about the limitation in Discussion section.
Point 1: (there is no line number)
Response 1: According to the reviewer’s suggestion, the line number was added in the revised manuscript.
Point 2: Full name of SPR should be described at the page 1 of 18 "In the last year, SPR-based biosensors ...."
Response 2: According to the reviewer’s suggestion, the full name of SPR was add in in the revised manuscript.
Point 3: The Table 1. "the dose of 102 (HAD50/ml)" --> "the dose of 102 HAD50/ml"
Response 3: According to the reviewer’s suggestion, we have removed the brackets in the revised manuscript.
Point 4: In page 11 of 18, only one word "Virotype" was italic. check it please.
Response 4: According to the reviewer’s suggestion, we have modified the text in the revised manuscript.
Point 5: In second line of Discussion and Conclusion section, wild swine --> wild boar
Response 5: According to the reviewer’s suggestion, we have modified the text in the revised manuscript.
Point 6: Still, I think that overall results are preliminary and difficult to apply in the field. Because the experiment was conducted 1) on cells only, and 2) it was not confirmed that the method could be distinguish other pathogens (PRRS, PCV2 etc.), and 3) only one virus was tested. In the discussion section, it should be described that these limitations (preliminary studies) and studies for future field application are needed.
Response 6: According to the reviewer’s suggestion, we have modified the text in the revised manuscript.
In the revised manuscript, we added the following sentence to the description to Discussion and Conclusion section: “Further studies are needed before to proceed with the development of an on-field application. The preliminary results obtained on the ASFV confirmed the possible application of SPR-based sensors for virus detection. In our case, to definitively confirm the performance of the assay, validation experiments on real swine samples (oral fluid, blood, or fecal samples) will be performed.”

Round 3
Reviewer 2 Report
Thank you for all changes in revision process.
Please delete " " in L480-L484